# The association of mental health with physical activity and its dimensions in Chinese adults: A cross-sectional study

Changming Shen[1], Yan Li [2]*

**1** Department of Physical Culture, Inner Mongolia University, Yuquan District, Hohhot City, Inner Mongolia Autonomous Region, China, **2** Department of Physical Education, Wenzhou University, Chashan Higher Education Park, Wenzhou City, Zhejiang Province, China

* tyxyliyan@wzu.edu.cn, Liyan885588@gmail.com

## Abstract

### Background

Selecting the most efficient type of physical activity that improves mental health can assist in choosing appropriate interventions. Hence, the objective of this study was to evaluate the associations between physical activity and its various aspects, including weekly physical activity, weekly walking and exercise sessions, and the frequency of walking and exercise per week, with the mental health of Chinese adults.

### Methods

This cross-sectional study was carried out in Hangzhou (2023) involving 512 adults aged 18 and 64. Each participant received a self-completed questionnaire comprising three sections. The initial section focused on gathering basic information about the participants, such as gender, age, annual income, and marital status. The second section consisted of the 12-item General Health Questionnaire (GHQ-12), which aimed to evaluate the mental health status of the participants. Lastly, the third section included the International Physical Activity Questionnaire–Short Version (IPAQ-SV), which assessed the metabolic equivalent (MET) of activities like walking, moderate-intensity exercises, and high-intensity exercises.

### Results

The study found that mental health problems affected 25.74% of adults, while physical inactivity was prevalent in 49.63% of adults. The statistical model was highly significant (F = 25.143, p < 0.001), suggesting that at least one predictor has a significant impact on mental health. The model accounted for 39% of the variance in mental health, with all variables showing predictive value. Notably, the number of walking days per week emerged as the most influential predictor of mental health (β = -0.392), followed by level of weekly physical activity in MET, the number of exercise training sessions per week, weekly exercise training in MET, and weekly walking in MET (β = -0.312, -0.301, -0.212, and -0.202, respectively).

**Data Availability Statement:** All relevant data are within the manuscript.

**Funding:** The author(s) received no specific funding for this work.

**Competing interests:** The authors have declared
that no competing interests exist.

## Conclusions

Adults can allocate more days per week to walking and their usual physical activity to
improve their mental health.

## Introduction

Mental health as a basic human right is an integral part of our well-being and general health.
People with good mental health are better able to function, connect, cope, and thrive. More-
over, mental health problems are the bedrock of many countries. Hence, it approximately
turns out that one in eight people on world have to cope with mental condition.

Also, a substantial proportion of the worldwide disease burden is related to mental health
conditions [1,2]. Overall, the economic consequences of mental health conditions and their
productivity losses and other indirect costs to society are much higher than health care costs.
In addition to, individuals with mental health conditions have a higher risk of non-communi-
cable diseases [1,3]. Over the last two decades (before COVID-19 pandemic), mental health in
China has become a focal point for both the general public and academic circles. The initial
national study on mental disorders in China, conducted from 2013 to 2015, revealed that
16.6% of adults have experienced mental health issues at some point in their lives [4]. Given
China's vast population, the demand for clinical intervention is substantial. In China and
many countries of the world, one of the problems that had a negative impact on mental health
was the COVID-19 virus and the pandemic caused by it. Many studies have shown that psy-
chological problems, including depression and stress increased during the pandemic in China
[5–7]. In the first national survey conducted in 2020, it was reported that 35% of respondents
experienced depression and anxiety during the first year of the pandemic [4]. Although the
measures taken by the Chinese government and many other governments to reduce the spread
of COVID-19 (including quarantine, closing schools, staying at home, etc.) were effective,
these measures can cause other problems, including the increase in mental problems. There-
fore, governments should prevent these problems by implementing preventive programs as
soon as possible.

Lifestyle can affect mental health [8]. One of the dimensions of lifestyle that is related to
mental health is physical activity [9]. Physical activity can affect mental health through differ-
ent mechanisms. For example, physical activity can increase self-esteem, reduce symptoms of
depression, anxiety and stress and increase cognitive function [10]. Also, physical activity can
increase the production of two neurochemicals (opioids and endocannabinoids) that are effec-
tive in reducing pain, improving mood and improving sleep [11]. However, the level of physi-
cal inactivity has increased compared to the past. It was observed that the percentage of adults
in China who didn't get enough physical activity grew from 17.9% in 2010 to 18.75% in 2022
[12,13].

Despite the numerous reports about the positive effects of regular physical activity on men-
tal health, there is still no detailed information about the best physical activity intervention
that causes positive effects on mental health. Physical activity encompasses various forms
including exercise training and active transport [14]. Each type of physical activity can result
in different outcomes [15,16].

Hence, the identification of the most effective form of physical activity that enhances men-
tal health can aid in selecting suitable interventions. Consequently, the aim of this study was to
assess the correlations between physical activity and its dimensions, such as the total weekly

physical activity, weekly walking and exercise training, and number of days of walking and exercise per week, with mental health among Chinese adults.

## Methods

### Study design and participants

This cross-sectional study was conducted in Hangzhou (April to August 2023), the capital and most populous city of Zhejiang, China, with a population of approximately 12 million and 10 counties (China Center for Population Statistics). First, 33 regions from 15 districts in Hangzhou were selected based on the number of health centers. In the next step, one health center was randomly selected from each region using the cluster sampling method. Adults aged 18 to 64 referring to selected health centers were considered as research samples. Participants with severe mobility limitations were excluded from the study. In addition, all participants who had one or more missing values for a variable were also excluded. A G-power was used to estimate the sample size. Based on the previous study [17] and taking into account $\alpha = 0.05$ and a power of 85%, the number determined was 319 participants. Due to the use of cluster sampling and a design effect of 1.5, the number of estimated samples increased to 479 participants. Given the possibility of losing 5% of the sample, the final sample size was ultimately increased to 503 participants. Each participant received a self-completed questionnaire that consisted of three parts. The first part covered the basic characteristics of participants, including gender, age, annual income, and marital status. The second and third parts were the GHQ-12 and IPAQ-SV questionnaires. This study was performed in line with the principles of the Declaration of Helsinki. Approval was granted by the Ethics Review of Wenzhou University (No. WZU2023024). The written consent was obtained from the participants to participate in the study.

### Mental health

The GHQ-12 reflects participants' mental health status using 12 self-assessment items. Each item includes four options (A, B, C, and D), and the bimodal scoring method (0-0-1-1) was adopted. Specifically, if A or B was selected, the value equals 0; If C or D was selected, the score equals 1. The mental health screening rate was considered positive (e.g., depression and anxiety, or insomnia) if the cumulative score was $\geq 4$ [18]. A higher score indicates a worse mental health condition. The Chinese version of GHQ-12 for professional groups [19] and general population [20] had high internal consistency. In this study, Cronbach's $\alpha$ coefficient for this scale was equal to 0.85, which indicates its good reliability.

### Physical activity

The IPAQ-SV assesses the metabolic equivalent (MET) of walking, moderate-intensity activities, and high-intensity activities. For each of these activities there is a metabolic equivalent [21]. Weekly physical activity level was calculated by summing the MET of the three types of physical activity. A participant was considered physically inactive if he/she did not meet any of these criteria:

 a) 3 or more days of vigorous activity of at least 20 minutes per day.

 OR

 b) 5 or more days of moderate intensity activity and/or walking of at least 30 minutes per day.

 OR

c) 5 or more days of any combination of walking, moderate-intensity, or vigorous-intensity activities to achieve a total PA of at least 600 MET minutes/week [21].

In addition to MET of total physical activity, the number of walking days per week, the number of exercise training sessions per week, and MET of walking and exercise training were also independently considered as predictors of mental health. Reliability for this scale in Chinese adults has been established (correlation coefficients above 0.70) [22]. The Cronbach's α coefficient of scale in the current study was 0.83, indicating good reliability.

## Statistical methods

The categorical variables of the study are presented in frequency (percentage) and the continuous variables of the study were presented in mean ± standard deviation. The difference between women and men in all variables was measured using the independent t-test. The linear multiple regression was used to assess the association between mental health score as a dependent variable and total weekly physical activity, weekly walking and exercise training, and number of days of walking and exercise per week as independent variables. The statistical software used was the Statistical Package for the Social Sciences (SPSS, IBM Corp., Armonk, NY, USA) version 26.0 ($p<0.05$).

## Results

In first, 592 participants participated in the study. However, 512 participants filled the questions correctly. In the sample included in these analyses, the mean age was 32.13±7.52 years, 317 (61.91%) were women, 420 (82.03%) adults were married, and 55 (10.74%) adults had an annual income of less than 100,000 Yuan (Table 1). The mental health problems and physical inactivity were prevalent in 26% and 49% of adults, respectively. There was no difference between female and male in the all variables ($p>0.05$).

Scatterplots of these variables against mental health revealed linear trends (VIF = 1.009). The Shapiro-Wilk test confirmed the residuals' normality (W = 0.97, p = 0.16). Homoscedasticity was confirmed, with a Breusch-Pagan test result of $\chi^2$ = 5.41, p = 0.11. The Durbin-Watson statistic of 1.96 suggests no auto-correlation, indicating independent errors. The model was significant (F = 25.143, p < 0.001), indicating at least one predictor significantly affects

**Table 1. Characteristics of the participants.**

| | | All = 512 | Female = 317,61.91% | Male = 195,38.09% | p |
|---|---|---|---|---|---|
| Age (year) | | 32.13±6.5 | 31.85±5.29 | 32.4±7.725 | 0.373 |
| Marital status | Married | 420 (82.03%) | 269 (84.85%) | 151 (77.435%) | |
| | Unmarried | 92 (17.97%) | 48 (15.16%) | 44 (22.565%) | |
| Annual income (Yuan) | <100,000 | 55 (10.74%) | 38 (11.987%) | 17 (8.72%) | |
| | 100,000 to 300,000 | 290 (57)% | 206 (64.99%) | 84 (43.07%) | |
| | 300,000 to 500,000 | 107 (21%) | 50 (15.78%) | 57 (29.23%) | |
| | 500,000 to 1,000,000 | 40 (7.33%) | 18 (5.68%) | 22 (11.29%) | |
| | >1,000,000 | 20 (4%) | 5 (1.57%) | 15 (7.69%) | |
| Mental health score | | 2.49±1.466 | 2.478±1.451 | 2.502±1.482 | 0.212 |
| Weekly total physical activity (MET) | | 643.22±495.19 | 640.32±485.95 | 646.12±501.43 | 0.290 |
| Number of walking days per week | | 2.21±2.59 | 2.26±2.52 | 2.16±2.66 | 0.091 |
| Number of exercise training sessions per week | | 1.31±2.03 | 1.299±1.99 | 1.331±2.09 | 0.398 |
| Weekly walking (MET) | | 284.11±510.57 | 289.62±505.39 | 278.6±515.75 | 0.077 |
| Weekly exercise training (MET) | | 512.63±943.97 | 508.81±930.81 | 516.45±957.13 | 0.101 |

**Table 2. Effect of physical activity and its dimensions on the mental health.**

| Model | Coefficients [a] | | | | | | |
|---|---|---|---|---|---|---|---|
| | Adjusted coefficients | | Unadjusted coefficients | t | P | 95% confidence interval for B | |
| | B | Std.Error | Beta | | | Minimum | Maximum |
| (Constant) | 82.138 | 5.392 | 83.451 | 11.548 | 0.01 | 72.191 | 92.085 |
| Total weekly physical activity | -0.312 | 0.044 | -0.329 | -4.171 | 0.021 | -0.308 | -0.316 |
| Walking per week | -0.202 | 0.053 | -0.204 | -2.99 | 0.039 | -0.189 | -0.215 |
| Exercise training per week | -0.212 | 0.041 | -0.216 | -3.39 | 0.01 | -0.201 | -0.223 |
| Number of walking days per week | -0.392 | 0.063 | -0.403 | -4.919 | 0.001 | -0.381 | -0.403 |
| Number of exercise training sessions per week | -0.301 | 0.049 | -0.305 | -4.101 | 0.01 | -0.290 | -0.312 |

$F = 25.143$, $R^2 = 0.39$, $p < 0.0001$; a. Dependent Variable: Mental health.

mental health. The model explains 39% of the variance in mental health, with an adjusted $R^2$ of 0.39. All variables predicted mental health. However, the number of walking days per week was the strongest predictor of mental health ($\beta = -0.392$). Subsequently, total weekly physical activity, number of exercise training sessions per week, exercise training per week, and walking per week ($\beta = -0.312$, $-0.301$, $-0.212$, and $-0.202$, respectively) (Table 2).

## Discussion

This study investigated the association between mental health and physical activity and its dimensions among Chinese adults. Prior to the outbreak of the COVID-19 pandemic, the issue of mental health had been rising in prominence within both the public sphere and academic circles in China. Findings from the first nationwide survey on mental disorders, carried out from 2013 to 2015, indicated that mental health conditions had affected 16.6% of the adult population in China at some stage in their lives [23]. In this study, the prevalence of mental health issues was reported to be 25.74%. Although by 2023, many pandemic restrictions have been lifted, the high prevalence of mental health issues compared to before the pandemic period can be related to the restrictions and problems related to the pandemic. However, the prevalence of problems has decreased compared to the pandemic era. According to the first national survey on psychological distress in the COVID-19 epidemic in China, 35% of respondents experienced stress, anxiety, and depression [4]. With vaccination rates increasing and infection rates declining in many areas, restrictions have been reduced since the end of 2022, allowing for the return of social interactions. Reuniting with friends and family, resuming social activities, and engaging in community events can significantly boost mood and reduce feelings of loneliness and isolation.

According to the current study, physical inactivity prevalence was 49.63%. Zhou et al., (2022) has shown that more than 46% of Chines adults were considered physically inactive in 2018 (before pandemic), a number that rose to 67.2% in 2020 (during pandemic) [24]. The results of the current research showed that inactivity has decreased compared to the pandemic era and is close to the pre-pandemic era. However, it is still one of the main health challenges. The global health emergency caused by the COVID-19 pandemic has led to a unique crisis in public health, resulting in significant changes to daily life. To curb the transmission of the virus, strict measures such as social distancing and home quarantines have been enforced worldwide. The closure of indoor and outdoor facilities during the pandemic due to city-wide lockdowns has limited physical activity opportunities [25].

The relationship between mental health and physical activity has been one of the constant lifestyle challenges [26]. As the results of the present study showed, physical activity was

associated with mental health. Also, different dimensions of physical activity, including weekly exercise training and walking, as well as the days of walking and exercising per week, were significantly related to mental health.

A systematic review examined the impact of physical activity on mental health in China and found physical inactivity was linked to higher rates of anxiety, depression, sleep disturbances, and lower subjective well-being [27]. Physical activity has a notable impact on mental well-being and overall quality of life [28]. Current theories suggest that depression can arise from some problems such as physical inactivity [29]. Additionally, physical inactivity can contribute to feelings of anxiety and worsen mental health disorders [30]. The relationship between physical activity and mental health appears to be bidirectional. Engaging in physical activity such exercise training can reduce the risk of anxiety and depression symptoms, but these symptoms may also hinder an individual's ability to engage in physical activity [31]. A study by Da Silva et al. (2012) found that regular physical activity was associated with a lower likelihood of experiencing depressive symptoms. Conversely, participants with symptoms of anxiety and depression were more likely to fall short of the recommended levels of physical activity [32]. However, there are potential mechanisms that could explain the impact of physical activity on mood. These mechanisms may involve the release of endorphins, thermogenesis, the activation of the mTOR axis in specific brain regions, and neurotransmitters such as dopamine and serotonin [33].

The strongest predictor of mental health in the present study was the number of walking days per week. This indicates that individuals who engaged in walking more frequently throughout the week experienced improved mental health. Wang et al, (2022) reported that minutes of walking per week has a significantly associated with better overall mental health in adults [17]. One potential explanation for the outcomes of this study is that increasing the frequency of walking per week may lead to heightened social interactions, resulting in positive psychological effects for the individual, such as stress reduction [34]. Some studies have indicated that diverse social interactions can impact the mental well-being of individuals, regardless of their circumstances [35]. This influence is shaped by a combination of the individual's perceptions and their personal and relational situations [36]. Furthermore, some studies have shown the potential for social interaction-based interventions to enhance mental health outcomes [36,37]. One additional factor that could be cited to explain the findings of the current study is that individuals who engage in walking for a greater number of days per week (regardless of the duration and intensity) actually enhance their likelihood of adopting and maintaining a healthy and active lifestyle [38,39]. Some studies have demonstrated that a healthy and active lifestyle can lead to improved mental well-being [40,41].

This study has some limitations. For example, one such limitation is the reliance on self-reported data. While the model fits well, it's crucial to note that it does not imply causation, and external factors not included in the model may also affect mental health.

## Conclusions

In general, the current study expands our understanding of how physical activity and its dimensions are related to overall mental health in Chinese adults. Although the current study showed the relationship between physical activity and its dimensions with mental health, the number of walking days per week was the best predictor of mental health. In other words, adults who were engaged in walking more days a week (regardless of the time and intensity of these activities), had better mental health. Therefore, adults can allocate more days per week to walking, along with their usual physical activity, in order to improve their mental health.

## Acknowledgments

We thank all participants for participating in this study.

## Author Contributions

**Conceptualization:** Changming Shen, Yan Li.

**Data curation:** Changming Shen, Yan Li.

**Formal analysis:** Yan Li.

**Investigation:** Changming Shen, Yan Li.

**Methodology:** Changming Shen, Yan Li.

**Project administration:** Yan Li.

**Software:** Changming Shen.

**Supervision:** Yan Li.

**Validation:** Yan Li.

**Writing – original draft:** Changming Shen, Yan Li.

**Writing – review & editing:** Yan Li.

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
