## [Decision Letter · Decision Letter 0]

8 Sep 2024

PONE-D-24-25870The association of mental health with physical activity and its dimensions in Chinese adults: A cross-sectional studyPLOS ONE

Dear Dr. Li,

Thank you for submitting your manuscript to PLOS ONE. After careful consideration, we feel that it has merit but does not fully meet PLOS ONE’s publication criteria as it currently stands. Therefore, we invite you to submit a revised version of the manuscript that addresses the points raised during the review process.

We look forward to receiving your revised manuscript.

Kind regards,

Jindong Chang, Ph.D.

Academic Editor

PLOS ONE

Journal Requirements:

https://www.frontiersin.org/journals/psychiatry/articles/10.3389/fpsyt.2020.565291/full?fbclid=IwAR

In your revision ensure you cite all your sources (including your own works), and quote or rephrase any duplicated text outside the methods section. Further consideration is dependent on these concerns being addressed.

Reviewers' comments:

Reviewer's Responses to Questions

**Comments to the Author**

1. Is the manuscript technically sound, and do the data support the conclusions?

Reviewer #1: Yes

Reviewer #2: Yes

2. Has the statistical analysis been performed appropriately and rigorously? 

Reviewer #1: Yes

Reviewer #2: Yes

3. Have the authors made all data underlying the findings in their manuscript fully available?

Reviewer #1: Yes

Reviewer #2: Yes

4. Is the manuscript presented in an intelligible fashion and written in standard English?

Reviewer #1: Yes

Reviewer #2: Yes

5. Review Comments to the Author

Reviewer #1: 1- The site of data collection is not clear. Did you collect the data online or from the clients to the health centers? Please explain.

2- Did you use string or cluster sampling method in choosing health centers? If yes, you should mention. If not, then please delete this sentence: Due to the use of cluster sampling and a design effect of 1.5 (109-110 lines).

3- Regarding the mental health and physical activity questionnaires, did you measure their reliability with Cronbach's alpha? (125-126 lines and 141-142 lines).

4- There was no difference between female and male in the all variables (p>0.05): You have only mentioned the study of gender differences (157-158 lines). However, in Table 1, there is no data related to each gender and the difference between them. Either add this difference to the table or remove the difference from the text altogether and enter gender as a predictor variable in the regression.

5- There is no need to mention the purpose of the research in the results section (161-164 lines).

Reviewer #2: This is an interesting topic area. However, I think there are a few issues that the authors may wish to consider.

Abstract: Modify lines 45 to 48 as follows: Notably, the number of walking days per week emerged as the most influential predictor of mental health (β= -0.392), followed by level of weekly physical activity in MET, the number of exercise training sessions per week, weekly exercise training in MET, and weekly walking in MET (β= -0.312, -0.301, -0.212, and -0.202, respectively).

Introduction: Provide more detailed reports of the prevalence of mental health problems in China. Furthermore, you have not provided any reports of physical inactivity in China.

Methods: The selection model of health centers is a cluster sampling. Please mention it in the sampling section.

Overall, how many health centers did you choose?

The method of using G power is correctly indicated.

Did you code or categorize the income level variable? Or have you used the quantitative information of this variable?

Both questionnaires measure mental health and physical activity well, so they are valid scales. However, did you calculated the reliability of the questionnaires with Cronbach's coefficient?

Please indicate which t-test you used.

Results: Please put the results or p value of t-test in table 1.

Please remove lines 161-163.

Please add Unstandardized coefficients and 95% confidence interval for B in table 2.

6. PLOS authors have the option to publish the peer review history of their article (what does this mean?). If published, this will include your full peer review and any attached files.

Reviewer #1: No

Reviewer #2: **Yes: **Hamid Amini

---

## [Author Response · Author response to Decision Letter 0]

17 Sep 2024

Reviewer #1: 1- The site of data collection is not clear. Did you collect the data online or from the clients to the health centers? Please explain.

Answer: I added: lines 116-117. Data were collected in selected health centers, not online.

2- Did you use string or cluster sampling method in choosing health centers? If yes, you should mention. If not, then please delete this sentence: Due to the use of cluster sampling and a design effect of 1.5 (109-110 lines).

Answer: Yes, it was a cluster sampling. Line: 116

3- Regarding the mental health and physical activity questionnaires, did you measure their reliability with Cronbach's alpha? 

Answer: Yes, I had measured it, but because these questionnaires had been used several times before in various Chinese articles, I thought they were highly valid. However, I added Cronbach's coefficient. lines: (138-139 lines and 155-156 lines).

4- There was no difference between female and male in the all variables (p>0.05): You have only mentioned the study of gender differences (157-158 lines). However, in Table 1, there is no data related to each gender and the difference between them. Either add this difference to the table or remove the difference from the text altogether and enter gender as a predictor variable in the regression.

Answer: I added in Table 1

5- There is no need to mention the purpose of the research in the results section (161-164 lines).

Answer: I removed. 

Reviewer #2: 

1- Abstract: Modify lines 45 to 48 as follows: Notably, the number of walking days per week emerged as the most influential predictor of mental health (β= -0.392), followed by level of weekly physical activity in MET, the number of exercise training sessions per week, weekly exercise training in MET, and weekly walking in MET (β= -0.312, -0.301, -0.212, and -0.202, respectively).

Answer: thanks dear reviewer, I edit. Lines: 45-49

2- Introduction: Provide more detailed reports of the prevalence of mental health problems in China. Furthermore, you have not provided any reports of physical inactivity in China.

Answer: I added in introduction. 

3- Methods: The selection model of health centers is a cluster sampling. Please mention it in the sampling section.

Answer: I added.

4- Overall, how many health centers did you choose?

Answer: 33 centers. 

5- Did you code or categorize the income level variable? Or have you used the quantitative information of this variable?

Answer: It is specified in Table 1. Annual income (Yuan): <100,000- 100,000 to 300,000- 300,000 to 500,000- 500,000 to 1,000,000- >1,000,000

6- Both questionnaires measure mental health and physical activity well, so they are valid scales. However, did you calculated the reliability of the questionnaires with Cronbach's coefficient?

Answer: Yes, I had measured it, but because these questionnaires had been used several times before in various Chinese articles, I thought they were highly valid. However, I added Cronbach's coefficient. lines: (138-139 lines and 155-156 lines).

7- Please indicate which t-test you used. 

Answer: I added in line 160. 

8- Results: Please put the results or p value of t-test in table 1.

Answer: I added in Table 1

9- Please remove lines 161-163.

Answer: I removed. 

10- Please add Unstandardized coefficients and 95% confidence interval for B in table 2.

Answer: I edit table 2

---

## [Editor Report · Decision Letter 1]

20 Sep 2024

The association of mental health with physical activity and its dimensions in Chinese adults: A cross-sectional study

PONE-D-24-25870R1

Dear Dr. Li,

We’re pleased to inform you that your manuscript has been judged scientifically suitable for publication and will be formally accepted for publication once it meets all outstanding technical requirements.

Kind regards,

Jindong Chang, Ph.D.

Academic Editor

PLOS ONE

---

## [Editor Report · Acceptance letter]

27 Sep 2024

PONE-D-24-25870R1 

PLOS ONE

Dear Dr. Li, 

I'm pleased to inform you that your manuscript has been deemed suitable for publication in PLOS ONE. Congratulations! Your manuscript is now being handed over to our production team.

Kind regards, 

on behalf of

Dr. Jindong Chang 

Academic Editor

PLOS ONE